# Valorization of Rice-Bran and Corn-Flour Hydrolysates for Optimized Polyhydroxybutyrate Biosynthesis: Statistical Process Design and Structural Verification

**DOI:** 10.3390/polym17141904

**Published:** 2025-07-10

**Authors:** Gaurav Shrimali, Hardik Shah, Kashyap Thummar, Esha Rami, Rajeshkumar Chaudhari, Jens Ejbye Schmidt, Ajit Gangawane

**Affiliations:** 1Department of Life Sciences, Parul Institute of Applied Science, Parul University, Vadodara 391760, Gujarat, India; esha.rami82036@paruluniversity.ac.in (E.R.); ajitgangawane124@gmail.com (A.G.); 2GTU-School of Applied Sciences and Technology, Gujarat Technological University, Chandkheda, Ahmedabad 382424, Gujarat, India; asso_prof_rc@gtu.edu.in; 3Department of Microbiology, Mehsana Urban Institute of Sciences, Ganpat University, Ganpat-Vidyanagar 384012, Gujarat, India; hardik.microbiologist@gmail.com; 4GTU-School of Pharmacy, Gujarat Technological University, Gandhinagar 382028, Gujarat, India; kasu_patel5@yahoo.co.in; 5Department of Green Technology, Campusvej 55, University of Southern Denmark, 5230 Odense, Denmark; 6Faculty of Science, SSSC, Swaminarayan University, Kalol 382725, Gujarat, India

**Keywords:** *Bacillus bingmayongensis* GS2, fermentation optimization, biodegradable polymer, sustainable substrates, microbial bioplastic

## Abstract

The extensive environmental pollution caused by petroleum-based plastics highlights the urgent need for sustainable, economically viable alternatives. The practical challenge of enhancing polyhydroxybutyrate (PHB) production with cost-effective agro-industrial residues—rice-bran and corn-flour hydrolysates—has been demonstrated. *Bacillus bingmayongensis* GS2 was isolated from soil samples collected at the Pirana municipal landfill in Ahmedabad, India, and identified through VITEK-2 biochemical profiling and 16S rDNA sequencing (GenBank accession OQ749793). Initial screening for PHB accumulation was performed using Sudan Black B staining. Optimization via a sequential one-variable-at-a-time (OVAT) approach identified optimal cultivation conditions (36 h inoculum age, 37 °C, pH 7.0, 100 rpm agitation), resulting in a PHB yield of 2.77 g L^−1^ (66% DCW). Further refinement using a central composite response surface methodology (RSM)—varying rice-bran hydrolysate, corn-flour hydrolysate, peptone concentration, and initial pH—significantly improved the PHB yield to 3.18 g L^−1^(74% DCW), representing more than a threefold enhancement over unoptimized conditions. Structural validation using Fourier Transform Infrared spectroscopy (FTIR) and Proton Nuclear Magnetic Resonance spectroscopy (^1^H-NMR) confirmed the molecular integrity of the produced PHB. That *Bacillus bingmayongensis* GS2 effectively converts low-cost agro-industrial residues into high-value bioplastics has been demonstrated, indicating substantial industrial potential. Future work will focus on bioreactor scale-up, targeted metabolic-engineering strategies, and comprehensive sustainability evaluations, including life-cycle assessment.

## 1. Introduction

The global reliance on synthetic polymers continues to escalate, with annual output climbing from 1.5 million metric tons in 1950 to more than 400 million metric tons today [1,2]. Forecasts indicate that production could exceed 650 million metric tons by 2050 [3]. This intensive consumption of petroleum-based, non-biodegradable plastics has led to severe environmental challenges, further exacerbated by population growth and the rise in single-use plastics during the SARS-CoV-2 pandemic. Consequently, there is an urgent need for sustainable alternatives, driving increased interest in biodegradable polymers [4].

Biodegradable biopolymers offer a viable mitigation route. Polyhydroxybutyrate (PHB), accumulated intracellularly by numerous bacteria as a carbon- and energy-reserve under nutrient limitation [5], matches the mechanical performance of polypropylene yet is fully compostable [6]. Widespread adoption, however, is hindered by high manufacturing costs stemming from substrate price and fermentation complexity. Recent advances target cheaper feedstocks, more productive strains and refined process control to reduce unit cost [7].

Several microbes illustrate this potential: *Pseudomonas plecoglossicida* M9 and *Pseudomonas baetica* Y6 achieved PHB yields of 0.94 g L^−1^ (78.8% DCW) and 0.81 g L^−1^ (74.3%, DCW), respectively [8]. Previous studies on related *Bacillus* species indicate typical PHB yields ranging from 0.8 to 2.5 g/L under optimized conditions. Specifically, *Bacillus megaterium* SFK produced approximately 1.2 g/L PHB (75% DCW) using fruit and vegetable scraps, while optimized conditions for *Bacillus cereus* M5 yielded up to 2.5 g/L (74% DCW) [9,10]. Notably, PHB production by *Bacillus bingmayongensis* has not previously been reported, underscoring the significance and novelty of these optimized results.

In the present study, *Bacillus bingmayongensis* GS2, isolated from the Pirana municipal landfill in Ahmedabad, demonstrates robust growth on cost-effective agricultural residues, namely enzymatically hydrolyzed rice bran and corn flour. Initial optimization using a sequential one-variable-at-a-time (OVAT) approach achieved a PHB yield of 2.77 g/L(66% cell dry mass), representing approximately a 131% improvement compared to *B. megaterium* SFK and about 11% higher than *B. cereus* M5. Subsequent statistical optimization via central composite response surface methodology (RSM)—varying rice-bran hydrolysate, corn-flour hydrolysate, peptone concentration, and initial pH—further enhanced the PHB yield to 3.18 g/L (74% DCW), an overall increase of approximately 165% compared to *B. megaterium* SFK and of 27% compared to *B. cereus* M5. PHB biosynthesis involves a three-step enzymatic pathway: condensation of acetyl-CoA molecules by β-ketoacyl-CoA thiolase (*phb*A), NADPH-dependent reduction to hydroxybutyryl-CoA by acetoacetyl-CoA reductase (*phb*B), and polymerization into PHB by PHB synthase (*phb*C). These enzymes, encoded by operons, regulate polymer synthesis under nutrient stress conditions, facilitating intracellular PHB accumulation [11].

Efficient exploitation of PHB requires continuous discovery and screening of robust producers [12]. Morphological, biochemical (VITEK-2), and 16S rDNA analyses confirmed the taxonomic identity of *Bacillus bingmayongensis* GS2. Optimization studies included inoculum age, temperature, pH, agitation, and nutrient composition, with polymer characterization verified by spectroscopic analyses (FTIR, ^1^H-NMR). By integrating low-cost agro-industrial residues with statistically optimized fermentation, this study provides a scalable, sustainable alternative to petroleum-based plastics, demonstrating significant industrial applicability and laying the groundwork for future pilot-scale evaluations. To our knowledge, this is the first study reporting the use of amylase-treated rice-bran and corn-flour hydrolysates specifically with *Bacillus bingmayongensis* GS2. While various agro-residues have been tested with other bacterial species, this specific combination—paired with comprehensive optimization of parameters including inoculum age, inoculum concentration, incubation temperature, pH, and agitation—represents a novel, integrative approach. These optimized parameters collectively enhance PHB yield and offer insights valuable for industrial scalability.

## 2. Material and Methods

### 2.1. Soil Sampling and Bacterial Extraction Techniques

Soil samples were collected from four different sites within the Pirana municipal waste facility (coordinates: 22°58′54.4″ N, 72°33′55.5″ E) located in Ahmedabad, Gujarat, India, to isolate bacteria capable of producing polyhydroxybutyrate (PHB). For each sample, 1 g of soil was suspended in 9 mL of sterile distilled water and subjected for incubation at 120 rpm for 30 min at 37 °C. Subsequently, serial dilutions were prepared, and aliquots were plated onto nutrient agar. The plates were incubated at 37 °C, for 24 h to allow bacterial colony development. Pure colonies were then isolated and preserved on nutrient agar slants at 4 °C, for future analyses [2].

### 2.2. Screening of PHB-Accumulating Bacterial Isolates Using Sudan Black B Staining

To detect potential PHB-producing bacteria, an initial screening was performed on Petri dishes using Sudan Black B staining. Bacterial smears were treated with a 0.3% alcoholic Sudan Black B solution for 20 min. The excess dye was then removed by washing with 96% ethanol, after which the dishes were visually inspected [13]. Colonies that exhibited a dark blue-black color were selected as potential PHB producers. These selected colonies were then subjected to further analysis on glass slides. Heat-fixed bacterial smears were stained using a 30% Sudan Black B solution, prepared by dissolving 0.3 mg of dye in 100 mL of 70% ethanol [14]. After a 15 min staining period, the slides were rinsed with xylene and subsequently counterstained with a 5% Safranin solution for 1 min [15]. The bacterial specimens were examined under a microscope using immersion oil at 100× magnification. Cells displaying blue-black cytoplasmic granules were identified as PHB producers and subsequently preserved in 2% glycerol vials to facilitate further analysis and ensure long-term storage.

### 2.3. Morphology and Biochemical Characterization of the Isolates

The morphological and biochemical characterization of the highest PHB producing bacteria was conducted in accordance with Bergey’s Manual of Systematic Bacteriology [16]. This comprehensive evaluation included detailed observation of colony and cellular morphology. Additionally, a series of biochemical tests were performed to assess the metabolic capabilities of the bacteria. The VITEK-2 compact system was employed for biochemical characterization and identification, delivering a 40-parameter physiological profile including carbon utilization and antibiotic sensitivity [17].

### 2.4. Molecular Identification and GenBank Accession Submission

Molecular identification of the isolates was performed by sequencing the 16S rDNA gene. Genomic DNA was extracted using a commercial kit (HipurA Bacterial genomic DNA Purification kit Himedia, Thane, India) and assessed by agarose gel electrophoresis and spectrophotometry (260/280 nm). The 16S rDNA region was amplified via PCR with universal primers under standard cycling conditions. PCR products were visualized on a 1% agarose gel and subsequently sequenced [18]. Sequence data were analyzed using BLAST (version 1.4.0) against the NCBI database for similarity searches and annotated with SeqScape software v4.0 to confirm bacterial identity [19,20].

### 2.5. Production, Extraction, and Quantification of PHB

PHB production by *Bacillus bingmayongensis* GS2 was carried according to a modified protocol based on [21]. The medium contains glucose as the primary carbon source along with essential salts and trace elements. The culture was incubated at 37 °C, 120 rpm for 48 h. For PHB extraction, cell biomass was harvested by centrifugation (8000 rpm, 15 min). Cells were disrupted chemically by sodium hypochlorite (5%) digestion for 2 h at room temperature, followed by repeated washing with distilled water, acetone, and methanol to remove residual impurities. Finally, purified PHB was obtained by dissolving the extracted polymer in boiling chloroform, filtered through Whatman filter paper, and recovered by evaporation. This methodology closely simulates scalable industrial processing conditions.

### 2.6. Polymer Characterization Using FTIR and NMR Analysis

The ^1^H NMR spectra of PHB were acquired at 100.71 MHz on an NMReady 100 spectrometer (Nanalysis Corp., Calgary, AB, Canada) using chloroform-d as solvent, with chemical shifts (δ) referenced against the solvent residual peak. The spectral acquisition was performed with 16 scans, 5.1 s relaxation delay, and 4.1 s acquisition time. The relaxation delay was set to 5.0759 s, and the acquisition time was 4.0670 s, providing comprehensive resolution and accurate peak identification. A spectral width of 2014.0 Hz was selected to encompass the full proton chemical shift range pertinent to PHB analysis.

### 2.7. Optimization of Culture Conditions for PHB Production by OVAT

PHB yield was optimized by individually varying culture parameters using OVAT [22,23]. This approach involved individually varying each parameter while keeping others constant to determine its specific impact on PHB yield. Inoculum age was tested at intervals of 18, 24, 36, 48, 60, and 72 h, while inoculum size was varied at 0.5%, 1%, 1.5%, 2%, 2.5%, and 3%. Incubation time was studied at 12 h intervals ranging from 12 to 96 h. The incubation temperature was examined at 28 °C, 37 °C, 40 °C, 45 °C, 50 °C, 55 °C, and 60 °C. Medium pH was adjusted across a range of 6 to 9 with intervals of 0.5 units. Agitation rates were tested at 50, 100, 150, 200, and 250 rpm to determine their impact on PHB synthesis. A variety of carbon sources, including glucose, galactose, maltose, ribose, sucrose, and fructose, were individually tested to evaluate their influence on PHB production. Various nitrogen sources, including peptone, beef extract, yeast extract, casein hydrolysate, urea, and tryptone, were evaluated individually to identify their specific roles in enhancing PHB production and bacterial growth.

### 2.8. Experimental Design for Medium Optimization Using Response Surface Methodology (RSM)

A central composite design (CCD) was employed to optimize the medium composition and assess the interactive effects of four independent variables: Rice Bran Treated with Amylase (RBA), Corn Flour Treated with Amylase (CFA), Peptone, and initial pH [14,24]. Rice-bran and corn-flour substrates were hydrolyzed enzymatically using commercial α-amylase. Substrates were individually suspended in distilled water at 5% (*w*/*v*), and the pH was adjusted to 6.0. α-Amylase (1% *v*/*w* substrate) was added, and hydrolysis was conducted at 55 °C under continuous agitation for 4 h. The enzymatic reaction was terminated by heating the mixture at 100 °C for 10 min, followed by rapid cooling. The hydrolysates were centrifuged (8000 rpm, 10 min) to obtain clarified supernatants for subsequent microbial fermentation [25]. CCD was applied using Design-Expert software 12.0 (Stat-Ease, Inc., Minneapolis, MN, USA) to systematically evaluate factor interactions across a predetermined range. The coded and actual levels of these parameters are detailed in Table 1, ensuring experimental consistency and accurate interpretation of responses.

A two-level, four-factor CCD was implemented to optimize PHB production, considering the impact of key parameters (RBA, CFA, Peptone, and pH). A total of 30 experimental trials, including six central replicates, were generated using Design-Expert software to ensure robust statistical evaluation of variable interactions. Experiments were conducted in 250 mL Erlenmeyer flasks containing 100 mL medium inoculated with 10 mL seed culture. Cultures were incubated at 37 °C and agitated at 150 rpm to ensure sufficient aeration. PHB yield was assessed by measuring dry cell weight (g/L) after 48 h of incubation. To assess the impact of variables, response surface plots (3D) were generated, providing a graphical representation of factor interactions. Additionally, the point prediction tool in Design-Expert software was used to validate the experimental outcomes by comparing predicted and observed values, ensuring accuracy in optimizing conditions for maximum PHB production.

### 2.9. Statistical Analysis and Model Validation

Statistical analysis was performed using analysis of variance (ANOVA) and regression modeling with Design-Expert 12.0 (Stat-Ease Inc., Minneapolis, MN, USA). Significance was determined at a confidence level of 95% (*p* < 0.05). ANOVA provided F-values, *p*-values, and confidence intervals to assess the significance and interactions of independent variables, ensuring robust model validation. All experiments were conducted with three independent biological replicates, each comprising three technical replicates for accuracy and reproducibility. Experimental conditions and samples were randomized to minimize potential bias.

## 3. Results

### 3.1. Screening Soil-Derived Bacteria for Polyhydroxybutyrate (PHB) Production

From the soil samples analyzed, thirteen bacterial isolates were successfully obtained. The preliminary screening using Sudan Black B staining revealed that five of these isolates accumulated PHB granules. Among these, the isolate GS 2 demonstrated the highest PHB production, making it the most promising candidate for further detailed characterization and analysis.

### 3.2. Molecular Identification of High PHB-Producing Isolate GS2 by 16S rDNA Gene Sequencing

For accurate taxonomic identification of isolate GS2, genomic DNA extraction and PCR amplification of the 16S rDNA gene were performed. Sequence analysis revealed significant similarity with *Bacillus bingmayongensis* (Figure 1). The 16S rDNA gene sequence obtained from isolate GS2 was submitted to GenBank (accession number: OQ749793). Phylogenetic analysis using the neighbor-joining method in MEGA 11 revealed that GS2 clustered with closely related strains of *Bacillus bingmayongensis* and *Bacillus pseudomycoides*, thereby confirming its identification as *Bacillus bingmayongensis* GS2. These findings provide a clear genetic framework supporting the isolate’s taxonomic placement.

### 3.3. Biochemical Profiling of Bacillus bingmayongensis GS2 Using VITEK-2: Insights into Metabolic and Resistance Traits

The biochemical characterization of *Bacillus bingmayongensis* GS2 revealed significant insights into its metabolic capabilities and enzymatic activities (Table 2). The organism demonstrated the ability to ferment specific carbohydrates, including d-Galactose, d-Ribose, N-Acetyl-Glucosamine, d-Maltose, Saccharose, and d-Trehalose, highlighting its preference for certain sugar substrates. However, it was unable to metabolize others, such as d-Xylose, d-Sorbitol, Lactose, and sugar alcohols like d-Mannitol.

Enzymatic profiling further substantiated these findings, revealing positive activity for arginine dihydrolase 1, α-glucosidase, and leucine arylamidase. Conversely, the strain exhibited no activity for enzymes such as α-amylase, β-galactosidase, β-glucuronidase, and alanine arylamidase. Resistance profiling of *B. bingmayongensis* GS2 provided additional insights into its physiological properties. The strain exhibited resistance to Polymyxin B, Bacitracin, O/129, and Optochin, suggesting the presence of intrinsic defense mechanisms against these antimicrobial agents. Conversely, the organism was sensitive to Novobiocin, indicating susceptibility to this antibiotic.

### 3.4. Production Optimization

#### 3.4.1. Impact of Inoculum Age on PHB Production

The influence of inoculum age on PHB production revealed that the maximum yield of 2.77 g/L (66% DCW) was achieved with a 36 h-old inoculum (Figure 2A). Beyond this point, PHB production declined, with yields decreasing to 1.76 g/L at 48 h and further diminishing at 60 and 72 h. This decline is likely due to reduced metabolic activity as cells exit the logarithmic growth phase.

#### 3.4.2. Influence of Inoculum Size on PHB Production

The inoculum size significantly affected PHB production, with the highest yield of 2.23 g/L (50% DCW) observed at 2%. Both lower and higher inoculum sizes resulted in reduced PHB yields, with 1.77 g/L at 1.5%, 1.4 g/L at 1%, and 1.1 g/L at 0.5% (Figure 2B). Inoculum sizes of 2.5% and 3% further decreased the yield to 1.24 g/L and 1.12 g/L, respectively. These results highlight the importance of optimizing inoculum size to balance nutrient utilization and metabolic activity. Suboptimal yields at extremes likely result from inadequate or excessive competition for nutrients and oxygen.

#### 3.4.3. Influence of Incubation Time on PHB Production

Incubation time plays a crucial role in PHB production, with the highest yield of 2.75 g/L (64% DCW) observed at 48 h. At shorter incubation periods (12–24 h), PHB accumulation was relatively low, with values of 0.03 g/L and 0.77 g/L, respectively (Figure 2C). PHB production increased to 1.24 g/L at 36 h but began to decline after 48 h, with yields of 1.45 g/L, 1.04 g/L, 0.36 g/L, and 0.24 g/L at 60, 72, 84, and 96 h, respectively. These findings highlight the critical role of optimizing incubation time to achieve maximum PHB yields. The observed decline is attributed to nutrient depletion and intracellular polymer degradation at prolonged incubation.

#### 3.4.4. Influence of Incubation Temperature on PHB Production

Incubation temperature significantly influenced PHB production, with the highest yield of 1.97 g/L (47% DCW) observed at 37 °C. At lower temperatures, such as 28 °C, the yield decreased to 0.88 g/L, while a slight increase to 40 °C resulted in a marginally lower yield of 1.74 g/L. However, further temperature increases led to a sharp decline in PHB production, with yields dropping to 0.42 g/L, 0.1 g/L, and 0 g/L at 45 °C, 50 °C, and 55 °C, respectively (Figure 2D). No PHB production was observed at 60 °C, highlighting the detrimental effect of extreme temperatures on microbial metabolism. Yield reductions at non-optimal temperatures reflect decreased enzyme efficiency and metabolic instability.

#### 3.4.5. Influence of pH of Media on PHB Production

The pH of the production medium demonstrated a pronounced influence on PHB synthesis, with the maximum yield of 2.55 g/L (60% DCW) observed at pH 7 (Figure 2E). A slightly alkaline environment (pH 7.5) resulted in a reduced yield of 2.01 g/L, whereas both acidic (pH 6) and highly alkaline (pH 9) conditions led to significantly diminished production, yielding only 0.08 g/L and 0.09 g/L, respectively. Extremes in pH adversely affect microbial enzyme activity and membrane integrity, reducing metabolic efficiency.

#### 3.4.6. Influence of Agitation Rate on PHB Production

The agitation rate had a significant impact on PHB production, with the highest yield of 2.11 g/L (52% DCW) observed at 100 rpm. However, varying agitation conditions, including 50 rpm and 250 rpm, resulted in lower yields (0.7 g/L), while intermediate rates of 150 rpm and 200 rpm yielded 1.15 g/L and 0.8 g/L, respectively (Figure 2F). The optimal yield at 100 rpm suggests that this rate facilitated better oxygen transfer and nutrient mixing, which are crucial for microbial growth and PHB accumulation. Non-optimal agitation rates likely impaired oxygen transfer and caused mechanical stress on cells.

#### 3.4.7. Influence of Carbon Sources on PHB Production

The carbon source significantly influenced PHB production, with glucose yielding the highest concentration of 2.55 g/L (62% DCW). Other carbon sources also supported PHB synthesis, with maltose resulting in 1.47 g/L, sucrose yielding 1.88 g/L, and fructose producing 1.35 g/L. Galactose and ribose contributed lower yields of 1.23 g/L and 1.25 g/L, respectively (Figure 3A). These findings suggest that glucose is the most effective carbon source for PHB production, likely due to its rapid assimilation and efficient support of microbial growth.

#### 3.4.8. Influence of Nitrogen Source on PHB Production

The nitrogen source played a significant role in PHB production, with yeast extract yielding the highest concentration of 2.76 g/L (66% DCW). Peptone also supported substantial PHB synthesis, producing 2.45 g/L, while casein hydrolysate and tryptone resulted in 2.24 g/L and 2.13 g/L, respectively. Beef extract produced a slightly lower yield of 1.75 g/L, and urea was the least effective, yielding only 0.21 g/L (Figure 3B). These findings highlight yeast extract as the most efficient nitrogen source for PHB production, likely due to its nutrient-dense composition, which promotes optimal microbial growth and biopolymer synthesis.

### 3.5. Characterization of PHB by FTIR and NMR Analysis

FTIR spectroscopy confirmed the polymeric structure of extracted PHB with characteristic absorption peaks. C-H stretching vibrations were observed at 2970.41, 2867.74, 2844.92, and 2826.38 cm^−1^. CH_3_ bending vibrations appeared at 1455.97 and 1393.23 cm^−1^, while C-O-C stretching indicative of ester bonds was detected at 1150.80 cm^−1^. Additional C-O-C linkages were confirmed by peaks at 1015.33 and 1031.02 cm^−1^. Minor O-H stretching at 3709.09 and 3664.88 cm^−1^ likely represented residual moisture. Lower frequency peaks at 770.05, 454.90, and 420.68 cm^−1^ corresponded to CH_2_ rocking and skeletal vibrations [26], as shown in (Figure 4).

^1^H NMR spectroscopy in CDCl_3_ at 33 °C revealed characteristic PHB signals. The methyl (CH_3_) group appeared as a distinct peak around 1.2 ppm, confirming PHB presence [27]. A prominent singlet at 7.29 ppm corresponded to residual chloroform solvent. Methine (CH) and methylene (CH_2_) peaks, typically expected near 5.2 ppm and 2.4–2.6 ppm, respectively, showed reduced intensity, likely due to concentration limitations. Minor peaks between 3 and 4 ppm may represent trace impurities or solvent remnants (Figure 5).

### 3.6. Production Optimization by Response Surface Methodology (RSM) Approach

A comprehensive four-factor, two-level factorial design was utilized to evaluate polyhydroxybutyrate (PHB) production. This design incorporated six replicates at the central point and six axial points, as indicated in Table 3.

The second-order polynomial model describing PHB production is represented by the following equation:PHB (Y) = 3.18 + (0.0489 × A) + (0.0465 × B) + (0.0092 × C) + (0.3321 × D) + (12.49 × A^2^) + (11.79 × B^2^) + (13.29 × C^2^) + (9.87 × D^2^) + (0.0228 × AB) + 
(0.0438 × AC) + (0.4478 × AD) + (0.0510 × BC) + (0.2426 × BD) + (0.0203 × CD)(1)

In this model, Y represents the PHB dry weight (*w*/*v*), A corresponds to rice bran hydrolyzed with amylase (RBA) (*w*/*v*), B denotes corn flour hydrolyzed with amylase (CFA) (*w*/*v*), C stands for peptone (*w*/*v*), and D indicates the initial pH of the medium.

#### 3.6.1. Statistical Evaluation of the Model

The statistical strength of the model was evaluated through analysis of variance (ANOVA) using the F-test. The findings demonstrate that the model is significant at a 99% confidence level (*p* < 0.05), with *p*-values below 0.1000 indicating that the respective model terms substantially affect PHB production (Table 4). Notably, A^2^, B^2^, C^2^, and D^2^ emerged as key factors influencing PHB yield. *p*-values serve as indicators of each coefficient’s relative importance, helping to elucidate interactions among the independent variables. Lower *p*-values correspond to higher significance of the associated coefficient. The ANOVA further revealed an F-value of 23.32 for PHB production (*p* < 0.0001), indicating that the probability of observing this F-value due to random noise is just 0.01%. To assess the model’s adequacy, the “Adeq Precision” ratio—reflecting the signal-to-noise ratio—was determined. Since a value greater than 4 is considered desirable, the observed ratio of 13.355 indicates a signal for optimizing PHB production. The coefficient of determination (R^2^) was calculated as 0.9561, indicating that the model explains 95.61% of the variation in PHB yield. Given that R^2^ values approach 1.0 for strong predictive models [28], this result confirms the reliability of the current model.

#### 3.6.2. Effect of Nutrient Interactions on PHB Production

The interactive effects of nutrient components on PHB production were systematically analyzed using three-dimensional response surface plots, with two variables varied simultaneously while others remained constant at their central points (Figure 6). Response surface methodology (RSM) identified statistically significant interactions among rice-bran hydrolysate, corn-flour hydrolysate, peptone, and initial medium pH, each critically influencing PHB synthesis. Optimal conditions predicted by the RSM model—8.75 g/L rice-bran hydrolysate, 6.4 g/L corn-flour hydrolysate, 8.75 g/L peptone, and pH 7—achieved a maximum PHB yield of 3.18 g/L. This close correspondence between predicted and actual yields highlights the robustness and reliability of the statistical optimization strategy employed.

## 4. Discussion

### 4.1. Isolation and Identification of Bacillus bingmayongensis GS2 Demonstrate Specialized Metabolic Capabilities Suited for PHB Production

The successful isolation of PHB-producing bacteria from soil samples demonstrates the ubiquitous distribution of polyhydroxyalkanoate (PHA)-accumulating microorganisms in terrestrial environments. The identification of *Bacillus bingmayongensis* GS2 as a high PHB producer adds to the growing repertoire of *Bacillus* species capable of bioplastic synthesis. This finding is particularly significant as *Bacillus* species are generally recognized as safe (GRAS) organisms, making them attractive candidates for industrial biotechnology applications [29]. The phylogenetic clustering with *B. pseudomycoides* suggests shared metabolic capabilities within this bacterial group, potentially indicating conserved PHA biosynthetic pathways.

The comprehensive biochemical characterization of *Bacillus bingmayongensis* GS2 reveals a sophisticated metabolic architecture characterized by selective carbohydrate utilization patterns that reflect both ecological adaptation and biotechnological potential. The organism demonstrates preferential metabolism of specific carbohydrates including d-galactose, d-ribose, N-acetyl-glucosamine, d-maltose, saccharose, and d-trehalose, while exhibiting metabolic constraint toward alternative substrates such as d-xylose, d-sorbitol, lactose, and d-mannitol. This selective substrate utilization pattern indicates the presence of specialized enzymatic pathways for preferred carbon sources and the absence of metabolic machinery for less favored compounds, suggesting metabolic channeling mechanisms that align optimally with polyhydroxybutyrate (PHB) biosynthesis requirements. Such substrate-specific metabolic channeling demonstrates that preferred carbon sources are strategically directed toward polymer synthesis rather than general cellular metabolism, representing an evolutionary adaptation that enhances biopolymer accumulation efficiency.

Enzymatic profiling substantiates these metabolic preferences, revealing positive activity for arginine dihydrolase 1, α-glucosidase, and leucine arylamidase, which collectively underscore the strain’s capacity to hydrolyze specific amino acids and carbohydrate derivatives. Conversely, the absence of enzymatic activity for α-amylase, β-galactosidase, β-glucuronidase, and alanine arylamidase indicates metabolic specialization likely shaped by the organism’s ecological niche and optimized for specific biotechnological applications. The resistance profiling further elucidates the strain’s physiological robustness, demonstrating intrinsic defense mechanisms against antimicrobial agents including Polymyxin B, Bacitracin, O/129, and Optochin, while maintaining sensitivity to Novobiocin. These biochemical and resistance characteristics collectively reflect sophisticated adaptive strategies that position *B. bingmayongensis* GS2 as a metabolically specialized organism with significant potential for industrial applications, particularly in biopolymer production where substrate-specific metabolic channeling and enzymatic specialization converge to create optimal conditions for polyhydroxyalkanoate synthesis and accumulation [30].

### 4.2. Optimization of Culture Conditions Significantly Enhances PHB Productivity in Bacillus bingmayongensis GS2

#### 4.2.1. Temporal and Cultural Parameters

The optimization of inoculum age revealed that mid-logarithmic phase cells (36 h) provide optimal PHB production capacity. This finding aligns with established principles of microbial physiology, where cells in exponential growth phase exhibit maximum metabolic activity and enzyme expression levels [31]. The decline in PHB production with extended inoculum age likely reflects cellular aging, reduced metabolic vigor, and potential accumulation of inhibitory metabolites.

The inoculum size optimization (2% optimal) demonstrates the critical balance between cell density and nutrient availability. Insufficient inoculum leads to prolonged lag phases and suboptimal biomass generation, while excessive inoculum creates competition for nutrients and oxygen transfer limitations. Excessive inoculum concentrations may lead to metabolic stress, as reported by *Bacillus paranthracis* [32], where a 3% inoculum size led to a PHA production of only 0.647 g/L. This observation is consistent with previous studies on *Bacillus* species, where moderate inoculum concentrations facilitate optimal growth kinetics and metabolite production [33].

The temporal dynamics of PHB accumulation, with peak production at 48 h followed by decline, reflect the typical biphasic nature of PHA biosynthesis. Initial growth phase focuses on biomass accumulation, followed by polymer synthesis under nutrient limitation. The subsequent decline likely results from polymer degradation by intracellular depolymerases or cellular lysis [6]. Notably, recent studies using *Eichhornia crassipes* reported peak PHB production at 72 h, emphasizing the importance of precise timing to avoid diminished yields at ex-tended durations [34].

#### 4.2.2. Environmental Parameters

Temperature optimization revealed 37 °C as optimal for PHB production, consistent with the mesophilic nature of *Bacillus bingmayongensis* GS2. The sharp decline in production at elevated temperatures (>40 °C) reflects protein denaturation and membrane instability, while suboptimal temperatures reduce enzymatic activity and metabolic flux. This temperature sensitivity emphasizes the importance of precise thermal control in industrial PHB production processes [35]. Similarly, a study on *Bacillus flexus* reported maximum PHB production at approximately 35 °C, with a significant decline at temperatures above or below this optimum [36]. These findings emphasize the importance of maintaining an optimal incubation temperature, particularly around 35–37 °C, to maximize PHB yields.

The pH optimization results (pH 7.0 optimal) align with the physiological requirements of most *Bacillus* species. Extreme pH conditions disrupt cellular homeostasis, affecting enzyme activity, membrane integrity, and metabolic pathway efficiency. The narrow pH tolerance observed suggests the need for robust pH control systems in scaled production processes [37]. These findings are consistent with previous research on *Bacillus megaterium*, which reported optimal PHB accumulation at pH 8, while extreme pH levels adversely affected microbial growth and biosynthetic pathways. The results highlight the necessity of pH optimization as a critical parameter for achieving maximum efficiency in bioplastic production processes [38].

Agitation rate optimization (100 rpm optimal) reflects the balance between oxygen transfer and mechanical stress. Insufficient agitation limits oxygen availability for aerobic metabolism, while excessive agitation can damage cells and create foam formation issues. The moderate optimal agitation rate is typical for *Bacillus* fermentations and ensures adequate mixing without cellular damage [39]. In contrast, studies on *Bacillus cereus* 12GS have shown optimal PHB production at 200 rpm for 72 h, highlighting the variation in optimal agitation rates depending on the strain and experimental conditions [40].

#### 4.2.3. Nutritional Requirements

Carbon source evaluation demonstrated glucose superiority for PHB production, consistent with its role as a preferred substrate for most PHA-producing bacteria. Glucose enters central metabolism through well-established pathways, providing efficient carbon flux toward acetyl-CoA and ultimately PHB biosynthesis. The variable performance of alternative carbon sources reflects differences in uptake rates, metabolic processing efficiency, and integration with PHB biosynthetic pathways [41]. In another study, *Cupriavidus necator* demonstrated higher PHB yields, exceeding 4 g/L when glucose was used as the carbon source [42]. Nitrogen source optimization highlighted yeast extract as the most effective component, likely due to its complex nutritional composition including amino acids, vitamins, and growth factors. The poor performance of inorganic nitrogen sources like urea emphasizes the importance of organic nitrogen for optimal *Bacillus* growth and PHB production. This finding has practical implications for medium cost optimization in industrial applications [43]. In a study using *Pseudomonas* sp. AK-4, optimal PHA production of 0.92 g/L was achieved with 1% (NH_4_)_2_SO_4_ as a nitrogen source [44], suggesting that selecting the appropriate nitrogen source is crucial for maximizing biopolymer yields. Carbon and nitrogen source screening revealed glucose and yeast extract as optimal for PHB production, guiding the selection of agro-residue hydrolysates and peptone in RSM design. This translated baseline insights into practical substrate choices for industrially relevant optimization.

### 4.3. Structural Characterization Validates the Successful Biosynthesis and Purity of PHB Produced by Bacillus bingmayongensis GS2

FTIR analysis confirmed the successful biosynthesis of authentic PHB with characteristic ester bonds and methyl/methylene groups. The spectral profile matches established PHB standards, validating the polymer’s chemical structure and purity. The presence of minor O-H stretching likely represents residual moisture rather than structural modifications, indicating successful polymer extraction and purification [45]. ^1^H NMR analysis provided additional structural confirmation, though peak intensities suggest relatively low polymer concentrations. The characteristic methyl peak at 1.2 ppm definitively confirms PHB presence [27], while the reduced visibility of methine and methylene peaks may reflect concentration limitations or molecular weight effects [45]. Future studies should consider concentration optimization for enhanced spectral resolution.

### 4.4. Response Surface Methodology Effectively Optimizes PHB Production, Substantially Improving Biopolymer Yields for Industrial Applications

The successful application of response surface methodology (RSM) for polyhydroxybutyrate (PHB) production optimization demonstrates the power of statistical experimental design for biotechnology applications, with the high R^2^ value (0.9561) and significant F-value indicating robust model predictability essential for process scale-up and industrial implementation. The identification of quadratic terms as primary factors suggests non-linear relationships between variables, emphasizing the complexity of microbial metabolic responses that characterize biopolymer biosynthesis [46]. Elevated carbon levels expand the intracellular acetyl-CoA pool, thereby accelerating the PhaA → PhaB → PhaC route to PHB [47]. When nitrogen is limiting, the resulting C/N imbalance diverts flux from biomass to polymer storage [48]. pH and temperature further tune enzyme kinetics and membrane stability, modulating both growth and PHB yield. Finally, the physiological state set by inoculum age and size dictates lag duration and the timing of entry into the PHB-accumulation phase [49].

The maximum predicted yield of 3.18 g/L under optimized conditions represents a significant improvement over unoptimized production, demonstrating the value of systematic optimization approaches while establishing a competitive benchmark against other *Bacillus* species. This achievement is particularly noteworthy when compared to *Bacillus subtilis*, which produces only up to 0.8 g/L of PHB under optimized fermentation parameters, representing a nearly four-fold enhancement in productivity [50]. Furthermore, the effectiveness of RSM in optimizing PHB biosynthesis has been validated through comparable results in *Bacillus cereus* strain SH-02, which achieved a similar 3.1 g/L PHB yield under optimized conditions, thereby corroborating RSM as a powerful statistical tool for enhancing biopolymer production through precise modulation of key process parameters [35]. While this yield compares favorably with other *Bacillus* species reported in the literature, optimization of additional parameters including dissolved oxygen levels, feeding strategies, and bioreactor design configurations could potentially achieve even higher yields, suggesting that the full potential of RSM-guided bioprocess optimization remains to be explored [29].

## 5. Conclusions

This study establishes *Bacillus bingmayongensis* GS2, a novel isolate from a municipal solid waste site, as a robust microbial candidate for polyhydroxybutyrate (PHB) production under optimized cultivation conditions. The isolate was screened via Sudan Black B staining, taxonomically confirmed through 16S rDNA sequencing, and biochemically profiled using the VITEK-2 system, revealing a distinct enzymatic and metabolic landscape conducive to PHB biosynthesis. A two-tiered optimization strategy—comprising classical one-variable-at-a-time (OVAT) methods followed by multivariate Response Surface Methodology (RSM)—was employed to maximize PHB yield. Preliminary optimization identified a 36 h inoculum age, 2% inoculum size, 48 h incubation, 37 °C temperature, pH 7.0, 100 rpm agitation, glucose, and yeast extract as ideal parameters. RSM further enhanced production, achieving a maximum PHB concentration of 3.18 g/L. The integration of RSM clearly demonstrated its capability to optimize complex nutrient interactions efficiently, resulting in significantly enhanced PHB production compared to non-optimized conditions. This study sets a clear foundation for further scaling and economic feasibility assessment for industrial applications. Spectroscopic characterization using FTIR and ^1^H NMR confirmed the structural integrity and characteristic functional groups of the biopolymer. These findings underscore the strain’s industrial potential for sustainable PHB production. Despite its high yield and adaptability, challenges related to process scalability and cost remain. Future work should emphasize bioprocess intensification via fed-batch strategies, economical substrate integration, and downstream processing optimization. Additionally, metabolic engineering and full-scale life cycle assessment (LCA) are warranted to further enhance productivity and environmental performance in bioplastic manufacturing systems.

## Figures and Tables

**Figure 1 polymers-17-01904-f001:**
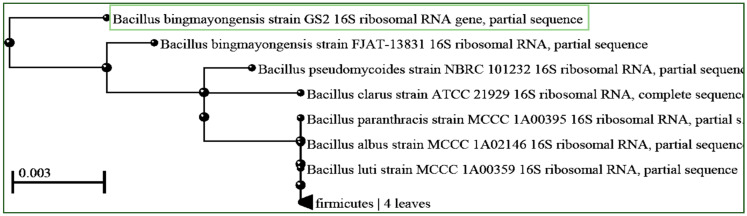
Phylogenetic tree of the genetic relationship of *Bacillus bingmayongensis* GS2 (showed in green box).

**Figure 2 polymers-17-01904-f002:**
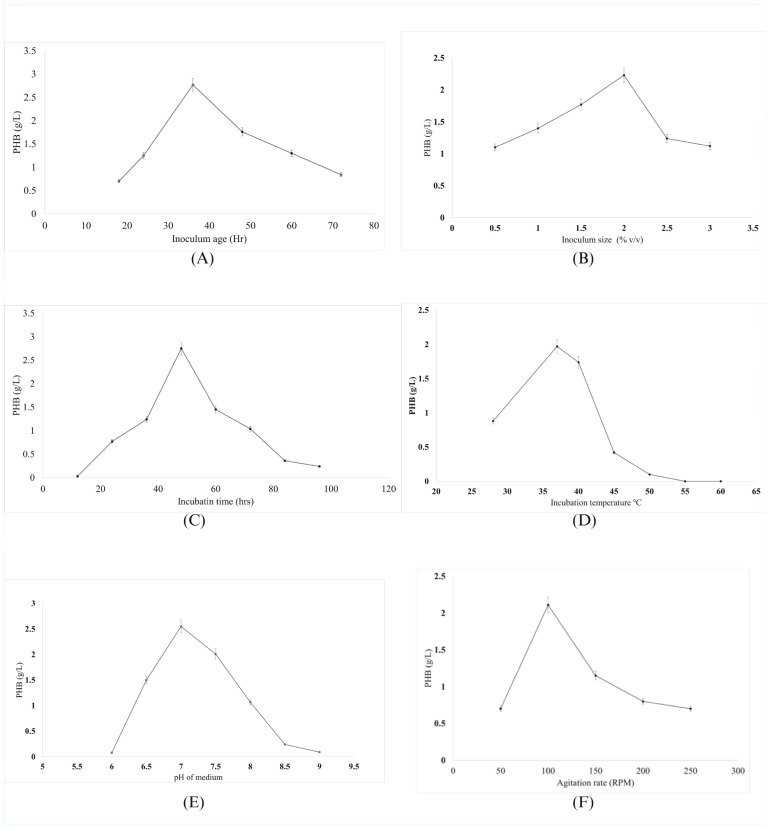
Effect of physical and cultural parameters on PHB production by *Bacillus bingmayongensis* GS2 (mean ± SD, *n* = 3; *p* < 0.05). (**A**) Inoculum age: peak PHB yield at 36 h. (**B**) Inoculum size: optimum at 2% *v*/*v*, with lower yields at both smaller and larger sizes. (**C**) Incubation time: maximum yield at 48 h, followed by a decline. (**D**) Incubation temperature: highest yield at 37 °C, with a sharp decline at higher temperatures. (**E**) Medium pH: optimal production at pH 7.0; reduced yields at acidic and alkaline conditions. (**F**) Agitation rate: maximum yield at 100 rpm, declining at both higher and lower speeds.

**Figure 3 polymers-17-01904-f003:**
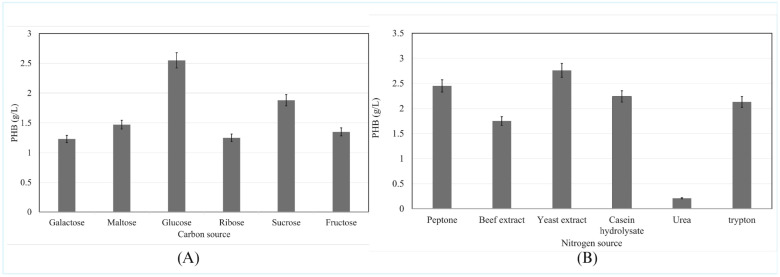
Effect of nutritional parameters on PHB production by *Bacillus bingmayongensis* GS2 (mean ± SD, *n* = 3; *p* < 0.05). (**A**) Effect of different carbon sources: glucose resulted in the highest PHB concentration compared to other tested sugars. (**B**) Effect of nitrogen sources: yeast extract produced the highest PHB concentration among the nitrogen sources evaluated.

**Figure 4 polymers-17-01904-f004:**
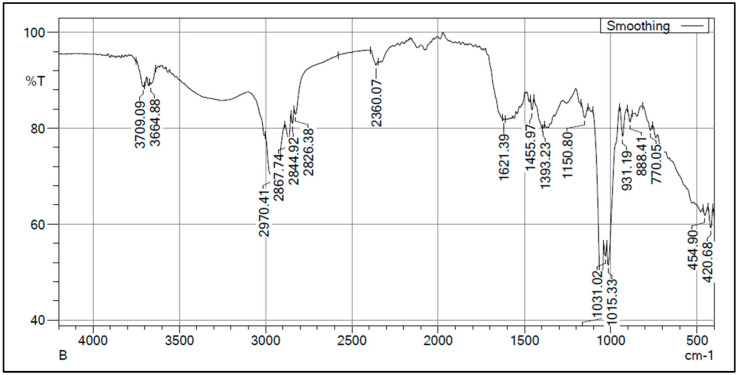
FTIR analysis of extracted PHB from *Bacillus bingmayongensis* GS2.

**Figure 5 polymers-17-01904-f005:**
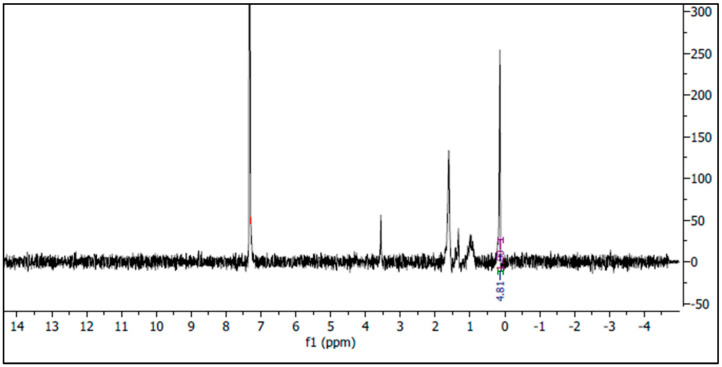
NMR analysis of extracted PHB from *Bacillus bingmayongensis* GS2.

**Figure 6 polymers-17-01904-f006:**
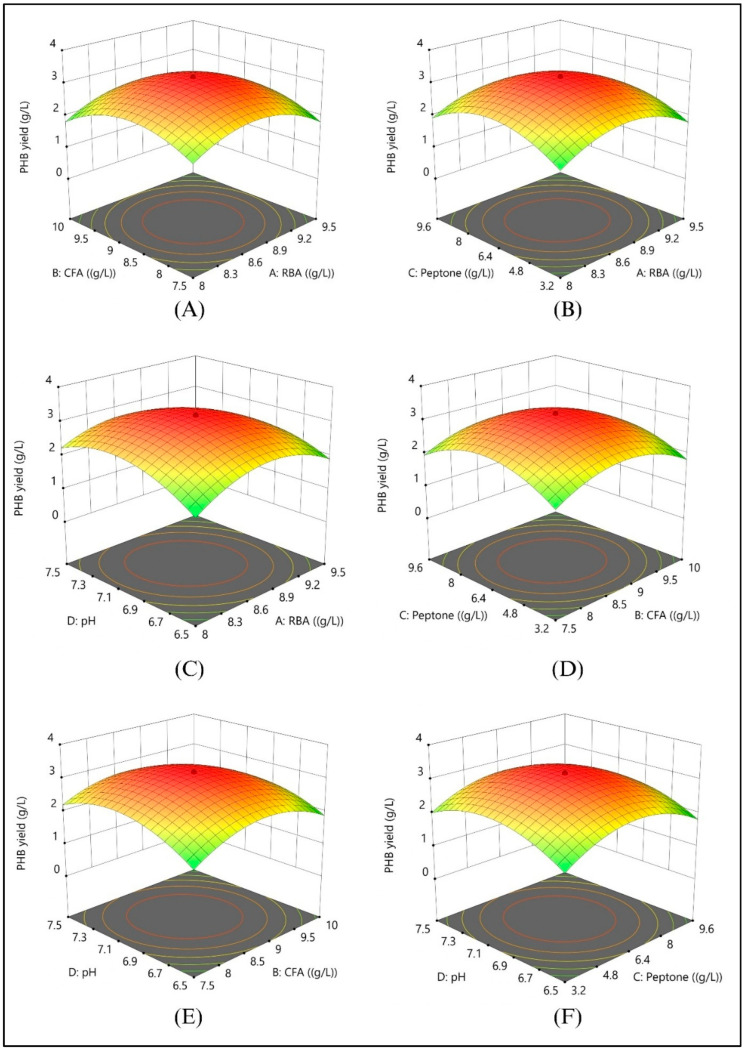
The figure represents three-dimensional response surface plots generated via Response Surface Methodology (RSM), illustrating the interactive effects of key variables on PHB yield (g/L). Specifically, these include (**A**) rice bran hydrolyzed with amylase (RBA) (*w*/*v*) and corn flour hydrolyzed with amylase (CFA) (*w*/*v*), (**B**) RBA (*w*/*v*) and peptone (*w*/*v*), (**C**) RBA (*w*/*v*) and initial pH, (**D**) CFA (*w*/*v*) and peptone (*w*/*v*), (**E**) CFA (*w*/*v*) and initial pH, and (**F**) peptone (*w*/*v*) and initial pH. The high coefficient of determination (R^2^ = 0.9831) and the very low *p*-value (<0.0001) underscore the robustness and statistical significance of the model’s predictive capability for PHB production.

**Table 1 polymers-17-01904-t001:** Coded levels of medium components for optimization.

Independent Variables	Symbols	Code Levels				
		−α	−1	0	+1	+α
RBA	A	7.25	8	8.75	9.5	10.25
CFA	B	6.25	7.5	8.75	10	11.25
Peptone	C	0	3.2	6.4	9.6	12.8
pH	D	6	6.5	7	7.5	8

±α values represent axial (extreme) levels; +1 and −1 values indicate the high and low factorial levels for each independent variable.

**Table 2 polymers-17-01904-t002:** Biochemical parameters of the isolate *Bacillus bingmayongensis* Strain GS2.

Sr No.	Test	Result (+/−)	Sr No	Test	Result (+/−)
1	Alpha-Amylase	−	22	d-Galactose Fermentation	+
2	Phosphatidylinositol Phospholipase C (PIPLC)	−	23	d-Ribose Fermentation	+
3	Arginine Dihydrolase 1	+	24	Lactose Fermentation	−
4	Beta-Galactosidase	−	25	N-Acetyl-Glucosamine Fermentation	+
5	Alpha-Glucosidase	+	26	d-Maltose Fermentation	+
6	Alkaline Phosphatase	−	27	d-Mannose Fermentation	−
7	L-Aspartate Arylamidase	−	28	d-Mannitol Fermentation	−
8	Beta-Galactosidase	−	29	Methyl-Beta-D-Glucopyranoside Fermentation	−
9	Alpha-Mannosidase	−	30	Pullulan Fermentation	−
10	Phosphatase	−	31	d-Raffinose Fermentation	−
11	Leucine Arylamidase	+	32	Salicin Fermentation	−
12	Proline Arylamidase	−	33	Saccharose Fermentation	+
13	Beta-Glucuronidase	−	34	d-Trehalose Fermentation	+
14	Alpha-Galactosidase	−	35	Urease	−
15	Pyroglutamyl Aminopeptidase	−	36	Polymyxin B Resistance	+
16	Beta-Glucuronidase	−	37	Bacitracin Resistance	+
17	Alanine Arylamidase	−	38	Novobiocin Resistance	−
18	Tyrosine Arylamidase	−	39	O/129 (2,4-diamino-6,7-diisopropylteridine) Resistance	+
19	Alcohol Dehydrogenase 2s	−	40	Optochin Resistance	+
20	d-Xylose Fermentation	−	41	L-Lactate Alkalinization	−
21	d-Sorbitol Fermentation	−	-	-	−

**Table 3 polymers-17-01904-t003:** Experimental design based on CCD and the result of the experiment.

Std	Run	RBA (g/L)	Peptone (g/L)	CFA(g/L)	pH	PHB Yield (g/L)
27	1	8.75	8.75	6.4	7	3.188
13	2	8	7.5	9.6	7.5	1.758
2	3	9.5	7.5	3.2	6.5	0.563
15	4	8	10	9.6	7.5	0.496
1	5	8	7.5	3.2	6.5	0.306
7	6	8	10	9.6	6.5	0.577
17	7	7.25	8.75	6.4	7	0.098
8	8	9.5	10	9.6	6.5	0.758
30	9	8.75	8.75	6.4	7	3.188
3	10	8	10	3.2	6.5	0.468
16	11	9.5	10	9.6	7.5	0.479
25	12	8.75	8.75	6.4	7	3.188
10	13	9.5	7.5	3.2	7.5	0.91
28	14	8.75	8.75	6.4	7	3.188
9	15	8	7.5	3.2	7.5	0.879
11	16	8	10	3.2	7.5	1.181
18	17	10.25	8.75	6.4	7	0.374
23	18	8.75	8.75	6.4	6	0.353
5	19	8	7.5	9.6	6.5	0.569
6	20	9.5	7.5	9.6	6.5	0.493
26	21	8.75	8.75	6.4	7	3.188
22	22	8.75	8.75	12.8	7	0.198
24	23	8.75	8.75	6.4	8	0.718
4	24	9.5	10	3.2	6.5	0.627
29	25	8.75	8.75	6.4	7	3.188
21	26	8.75	8.75	0	7	0.103
19	27	8.75	6.25	6.4	7	0.335
12	28	9.5	10	3.2	7.5	0.337
14	29	9.5	7.5	9.6	7.5	0.427
20	30	8.75	11.25	6.4	7	0.293

**Table 4 polymers-17-01904-t004:** Analysis of variance (ANOVA) for the regression model of PHB production based on experimental data.

Source	Sum of Squares	df	Mean Square	F-Value	*p*-Value
Model	34.55	14	2.47	23.32	<0.0001 significant
A-RBA	0.0489	1	0.0489	0.4621	0.5070
B-CFA	0.0465	1	0.0465	0.4397	0.5173
C-Peptone	0.0092	1	0.0092	0.0870	0.7721
D-pH	0.3321	1	0.3321	3.14	0.0968
AB	0.0228	1	0.0228	0.2150	0.6495
AC	0.0438	1	0.0438	0.4135	0.5299
AD	0.4478	1	0.4478	4.23	0.0575
BC	0.0510	1	0.0510	0.4820	0.4981
BD	0.2426	1	0.2426	2.29	0.1508
CD	0.0203	1	0.0203	0.1919	0.6676
A^2^	12.49	1	12.49	118.07	<0.0001
B^2^	11.79	1	11.79	111.46	<0.0001
C^2^	13.29	1	13.29	125.62	<0.0001
D^2^	9.87	1	9.87	93.29	<0.0001
Residual	1.59	15	0.1058		
Lack of Fit	1.59	10	0.1587		
Pure Error	0.0000	5	0.0000		
Cor Total	36.13	29			

## Data Availability

The raw data supporting the conclusions of this article will be made available by the authors upon request.

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
