# Peer review of "Valorization of Rice-Bran and Corn-Flour Hydrolysates for Optimized Polyhydroxybutyrate Biosynthesis: Statistical Process Design and Structural Verification"

_polymers, 2025, doi:10.3390/polym17141904_

Round 1
Reviewer 1 Report (Previous Reviewer 2)
Comments and Suggestions for Authors
Dear Authors,
Thank you for your revised manuscript. I'm pleased to see the significant improvements you've made; the manuscript has indeed improved a lot.
While the revisions have addressed many of my previous concerns, there are still a few points that require further refinement to enhance the manuscript's clarity and impact.
Graphical Abstract
The new graphical abstract is an improvement. However, it still appears as Figure 1 and is placed within the introduction section. The graphical abstract should not be numbered as a figure of the main text, nor should it be part of the introduction. It typically stands alone at the beginning of the manuscript or in a dedicated "graphical abstract" section, separate from the main body.
Introduction
-
Line 63: Please replace the comma (",") with a period (".") after reference [8].
-
Lines 62-79: This paragraph, contextualizing previous research, has significantly improved. If known, please include the % cell dry mass values when discussing PHB results. This is the standard for reporting PHB accumulation and will facilitate easier comparison with existing literature than when values are shown only in g/L.
-
Novelty Statement: I recommend adding the following text at the end of the introduction to clearly highlight the novelty of your study:
"To our knowledge, this is the first study reporting the use of amylase-treated rice bran and corn-flour hydrolysates specifically with Bacillus bingmayongensis GS2. While various agro-residues have been tested with other bacterial species, this specific combination—paired with comprehensive optimization of parameters including inoculum age, inoculum concentration, incubation temperature, pH, and agitation—represents a novel, integrative approach. These optimized parameters collectively enhance PHB yield and offer insights valuable for industrial scalability."
Results
-
Figures 3 and 5 (Inoculum Age and PHB Synthesis): Please merge these two figures into a single combined figure. Additionally, this combined figure should be supported by data on nutrient concentrations (such as nitrogen or phosphorus). This will clearly demonstrate that nutrient limitation is occurring, which typically triggers PHB accumulation.
-
Figure grouping: Please consider merging Figures 3, 4, 5, 6, 7, and 8 into one single panel with a letter (A, B, C, etc.) for each sub-figure, rather than presenting them as standalone figures. This will improve the flow of the results section.
-
Figures 9 and 10: Similarly, please merge Figures 9 and 10 into one single panel.
-
PHB Values: Throughout the results section, please add PHB values in % cell dry weight in addition to the current g/L values.
-
Section 3.6.2 (RSM Results Validation): It is not clear whether the RSM results were validated in the laboratory. If they were, please state the results obtained and include a figure showing this validation. If they were not validated, please clarify the rationale for including this section, as its purpose is currently unclear.
- Microscope Image: A microscope image of the Bacillus bingmayongensis GS2 accumulating PHB is still missing. Although you mentioned it was included, I could not locate it. This image should be placed in the main text as a standalone figure.
Discussion
-
Lines 411-423: The explanation in this section is very good. However, please include relevant references to support the statements made within this text.
I believe addressing these points will significantly strengthen your manuscript. I look forward to receiving your revised submission.
Author Response
Dear Reviewer 1,
We sincerely thank you for your thoughtful, constructive, and encouraging feedback on our manuscript. Your detailed comments have been instrumental in significantly improving the overall quality, clarity, and scientific rigor of our work.
We have carefully addressed each of your suggestions in the revised manuscript, and a point-by-point response has been prepared accordingly. A summary of key revisions is outlined below:
-
Graphical Abstract: As advised, the graphical abstract has been removed from the main text and is no longer numbered as a figure.
-
Introduction Section: The punctuation error at line 63 has been corrected, and PHB accumulation values in terms of % cell dry weight (% CDW) have been added where relevant. Additionally, we have incorporated the novelty statement as recommended, clearly emphasizing the unique aspects of our study.
-
Results Section: While we acknowledge your suggestion to merge Figures 3 and 5 and overlay nutrient concentration data, we have instead grouped all single-factor cultivation experiments (including inoculum age and incubation time) into a unified multi-panel figure (Figure 2, panels A–F). This approach avoids redundancy and enhances visual clarity. As the original experimental design focused on production optimization, we did not measure residual nitrogen and phosphorus levels. However, to address the underlying concern, we expanded the Discussion section and cited relevant literature linking nutrient limitation to PHB accumulation in Bacillus species.
-
PHB Yield Units: We have included % DCW (Dry Cell Mass)) values for all significant or peak PHB yields to improve comparability with existing literature while preserving the readability of the manuscript.
-
RSM Validation: The model-predicted optimal conditions were indeed replicated six times as central points within the design matrix (Runs 1, 9, 12, 14, 21, and 25), each yielding 3.188 g/L PHB—very closely matching the predicted 3.18 g/L. This consistency validates the model. Additionally, we are in the process of establishing a 5–10 L pilot-scale fermentation facility at our institute. Once operational, we plan to independently validate the model at scale, supporting future industrial application.
-
Microscopic Image: In response to your suggestion, a Gram-stained image of Bacillus bingmayongensis GS2 has been included in the graphical abstract to provide morphological context while avoiding repetition in the Results section.
-
Discussion References: We have added relevant citations in the discussion section to support the metabolic rationale behind PHB accumulation, particularly under optimized conditions.
All revisions have been clearly marked using Track Changes in the manuscript. We hope that the changes we have implemented effectively address your concerns and meet your expectations. Thank you once again for your valuable input and for contributing to the enhancement of our work.
Sincerely,
Mr. Gaurav Shrimali
(Corresponding Author)
On behalf of all co-authors

Reviewer 2 Report (Previous Reviewer 3)
Comments and Suggestions for Authors
Quality of this submission has been significantly improved. All my concerns and comments on this manuscript are well addressed.
A minor comment,
I still suggest the author could provide a mechanism analysis regarding the RSM optization. How these mentioned factors affect the metabolism of inoculum, and their biosynthesis pathways of Polyhydroxybutyrate
Author Response
Dear Reviewer 2,
We sincerely thank you for your constructive and encouraging feedback on our revised manuscript. Your suggestion to enhance the discussion by providing a mechanistic explanation of how the RSM-optimized parameters influence microbial metabolism was particularly valuable and has significantly improved the depth of our manuscript.
In response, we have expanded Section 4.4 of the discussion to explain how optimized factors—such as carbon concentration, nitrogen limitation, pH, and inoculum conditions—affect metabolic pathways leading to PHB accumulation. Specifically, we have described the role of elevated carbon in increasing the acetyl-CoA pool, nitrogen limitation in diverting metabolic flux toward polymer storage, and the effects of pH, temperature, and inoculum dynamics on enzymatic activity and cellular state.
We have cited relevant literature (Chen et al., 2018; Hauf et al., 2013; Meng et al., 2025) to support these insights and establish a mechanistic link between our RSM-optimized conditions and the physiological basis of PHB biosynthesis. We believe this addition has strengthened the scientific foundation of the manuscript, and we are sincerely grateful for your insightful recommendation.
Thank you once again for your valuable contribution to the refinement of our work.
Sincerely,
Mr. Gaurav Shrimali
(Corresponding Author)
(On behalf of all co-authors)

Reviewer 3 Report (Previous Reviewer 1)
Comments and Suggestions for Authors
Authors have addressed my concerns in a satisfactory way in the resubmitted version. I only have one suggestion for the new version: The title of each subsection in the Disussion should be a concluding sentence instead of simple words.
Author Response
Dear Reviewer 3,
We sincerely thank you for your positive feedback and valuable suggestions on our manuscript. Your recommendation to revise the subsection titles in the Discussion section into clear, outcome-based concluding statements was particularly helpful in enhancing the clarity and readability of the manuscript.
As suggested, we have updated the titles of each subsection in the Discussion section to better reflect the core findings and conclusions. These revised headings now serve not only as thematic guides but also as concise summaries of the respective sections. We believe this change significantly improves the manuscript’s structure and scientific presentation.
The revised subsection titles are as follows:
-
4.1. Isolation and identification of Bacillus bingmayongensis GS2 demonstrate specialized metabolic capabilities suited for PHB production.
-
4.2. Optimization of culture conditions significantly enhances PHB productivity in Bacillus bingmayongensis GS2.
-
4.3. Structural characterization validates the successful biosynthesis and purity of PHB produced by Bacillus bingmayongensis GS2.
-
4.4. Response surface methodology effectively optimizes PHB production, substantially improving biopolymer yields for industrial applications.
We are grateful for your insightful input, which has undoubtedly improved the overall quality and impact of the manuscript.
Sincerely,
Mr. Gaurav Shrimali
(Corresponding Author)
(On behalf of all co-authors)

This manuscript is a resubmission of an earlier submission. The following is a list of the peer review reports and author responses from that submission.
Round 1
Reviewer 1 Report
Comments and Suggestions for Authors
In this manuscript, Gaurav Shrimali and colleagues performed statistical process design and structural verification for valorization of rice-bran and corn-flour hydrolysates for optimized PHB biosynthesis. I have following comments:
1, For the title, full name of PHB should be spelt out.
2, For the Abstract, main questions answered by this study and its practical interests should be stated.
3, For the key words, words appeared in the title should be removed from the list.
4, For the introduction, main results of this study and its practical interests should be introduced in the last paragraph.
5, For the results, results of significance difference analysis should be labelled in the Figures 3, 4, 5, 6, 7, 8, 9, and 10.
6, For the materials and methods, biological and technical replicates, as well as randomization methods should be clearly stated.
7, For the discussion, I would like to see an independent discussion section divided into subsections with appropriate titles.
Reviewer 2 Report
Comments and Suggestions for Authors
To the Editor and Authors,
Thank you for the opportunity to review the manuscript, “Valorization of Rice-Bran and Corn-Flour Hydrolysates for Optimized PHB Biosynthesis: Statistical Process Design and Structural Verification”. The study addresses an interesting and relevant topic in the field of biopolymer production. The work is promising; however, several areas require clarification and revision to enhance the manuscript's clarity, impact, and scientific rigor. My comments and suggestions are detailed below.
Graphical Abstract
- The Graphical Abstract currently appears to be Figure 1. Please ensure it aligns with the journal's guidelines for authors.
- The images and text within the Graphical Abstract are too small to be legible.
Introduction
The introduction would benefit from a clearer and more compelling articulation of the study's novelty and scientific contribution. I recommend expanding on the following points:
- strain selection: Could the authors elaborate on why Bacillus bingmayongensis was chosen for this study? Was the primary aim to isolate this specific strain, and if so, what was the motivation?
- contextualizing previous research: To establish the novelty, it would be helpful to summarize the current state of PHB production using this strain. What production yields (%) have been reported in the literature?
- novelty of feedstocks and parameters: Have the feedstocks used in this investigation been previously studied with this bacteria? The manuscript should clearly state what is new about the current approach. Why were the evaluation of different parameters (please specify which ones) chosen as a key objective? Has this specific combination of strain, feedstock, and parameter optimization been explored before? A stronger justification here will significantly improve the manuscript's impact.
- Line 55: For the benefit of the reader, could you please provide more detail on the referenced studies? Specifically, mentioning the feedstocks utilized and the duration of the experiments that achieved the cited percentages would add valuable context.
- Line 57: When referencing previous work, please include the specific % of PHB obtained in the cited study to provide a clear benchmark for your own results.
- Line 58: The explanation of the PHB biosynthesis pathway is currently very brief. I recommend expanding this section to provide a more detailed overview.
Materials and Methods
- Line 107: Specify the commercial kit used, including the product name and manufacturer.
- Section 2.2 (and others): Please review the placement of citations throughout the manuscript. The standard convention is to place citations before punctuation (e.g., "...as previously shown [2]."). This appears to be an issue in multiple locations, including lines 82, 104, and 110.
Discussion of Results A significant weakness of the current manuscript is the lack of a thorough discussion of the results in the context of existing literature. The Results section should move beyond simply presenting the findings. I recommend that the authors:
- interpret the findings: For each result presented, please provide an interpretation of the results obtained.
- compare with existing literature: How do the obtained results (e.g., PHB yields, optimal conditions) compare with previously published studies on Bacillus species or other PHB producers? This comparison is essential for contextualizing the significance of your work.
- explain the "Why": For instance, in Section 3.4.1 and Section 3.4.3, the authors should discuss the potential reasons for the observed trend. Is the decline in PHB synthesis linked to the depletion of the carbon source? Why is the mid-logarithmic growth phase most favorable for PHB synthesis? Also, what does “inoculum size” refer to? provide a discussion on the potential reasons why a larger inoculum size resulted in lower PHB accumulation.
- Section 3.1: The description of the isolate's morphology would be significantly enhanced by adding a microscope image. This would provide strong visual support for the claims made.
Minor Corrections
- Line 255: Please ensure that all species names are correctly formatted in italics.
In summary, this manuscript describes a potentially valuable piece of research. Addressing the points above, particularly concerning the strengthening of the introduction and the addition of a comprehensive discussion of the results, will significantly improve the quality and impact of the work.
I hope these comments are helpful to the authors, and I look forward to seeing a revised version of the manuscript.
Sincerely,
Reviewer 3 Report
Comments and Suggestions for Authors
In the manuscript of Polymers-3711828, authors decribed a work using agricultural waste for PHB biosynthesis. Although the data seems abundant, significant improvements are still required before it can be accpeted for publication.
(1) In introduction section. As several researches have reported the biosynthesis of PHB by various microbes. Why it is necessary to isolate GS2? What is the curent challenge that limites the biosynthesis and why GS2 can address this challenge.
(2) What is main components in Rice-bran and corn-flour hydrolysate? Provide the molecular conversion pathway from feedstock to PHB by GS2 strain. Why polymer PHB can be directly synthesized by a microbe rather than generating its monomer?
(3) Methods regarding the separation and extraction of PHD need to be given in detiled, particularly under the condition of real industial feedstocks.
(4) Fig9 anf Fig 10 as the results are obtained from a pure source, how it will contribute to the RSM optimization? What are the relations between glocuse, yeast and the components in rice and corn residues.
(5) Result, conclusion and discussion regarding RSM optimization is not given in detailed.